# Improved Spring Vegetation Phenology Calculation Method Using a Coupled Model and Anomalous Point Detection

**Qian Luo [1,2], Jinling Song [1,2,*], Lei Yang [1,2] and Jindi Wang [1,2]** 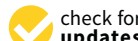

[1] State Key Laboratory of Remote Sensing Science, Jointly Sponsored by Beijing Normal University and Institute of Remote Sensing and Digital Earth of Chinese Academy of Sciences, Beijing 100875, China; luoqianzm@mail.bnu.edu.cn (Q.L.); yll2018@mail.bnu.edu.cn (L.Y.); wangjd@bnu.edu.cn (J.W.)

[2] Beijing Engineering Research Center for Global Land Remote Sensing Products, Institute of Remote Sensing Science and Engineering, Faculty of Geographical Science, Beijing Normal University, Beijing 100875, China

[*] Correspondence: songjl@bnu.edu.cn; Tel.: +86-10-5880-5452; Fax: +86-10-5880-5274

**Abstract:** High temporal resolution remote sensing satellite data can be used to collect vegetation phenology observations over regional and global scales. Logistic and polynomial functions are the most widely used methods for fitting time series normalized difference vegetation index (NDVI) derived from the Moderate Resolution Imaging Spectroradiometer (MODIS). Furthermore, the maximum in the curvature of the fitted curve is usually considered as the spring green-up date. However, the existing green-up date calculation methods have low accuracy for sparse vegetation. This paper proposes an improved green-up date calculation method using a coupled model and anomalous point detection (CMAPD). This model is based on a combination of logistic and polynomial functions, which is used to fit time series vegetation index. Anomalous values were identified using the nearest neighbor algorithm, and these values were corrected by the combination of growing degree-days (GDD) and land use type. Then, the trends and spatial patterns of green-up date was analyzed in the Sanjiangyuan area. The results show that the coupled model fit the time series data better than a single logistic or polynomial function. Besides, the anomalous point detection method properly controlled the green-up date within the local threshold, and could reflect green-up date more accurately. In addition, a weak statistically significant advance trend for average vegetation green-up date was observed from 2000 to 2016. However, in 10.4% of the study area, the the green-up date has significant advanced. Regression analysis showed that the green-up date is correlated to elevation: the green-up date is clearly later at higher elevations.

**Keywords:** MODIS NDVI; curve fitting; growing degree-days; green-up date; CMAPD; elevation gradient

## 1. Introduction

The scope of global warming has expanded to the ecological zone in China, and one of the most serious impacts is observed in the Sanjiangyuan Nature Reserve located in Qinghai Province. The Sanjiangyuan Nature Reserve is the largest nature reserve in China and an ecologically sensitive area at a high altitude [1]. Therefore, it is of great significance to monitor the ecological environment in this region. Climate warming can lead to changes in the ecological environment, which directly result in changes in vegetation productivity, biodiversity, ecosystem carbon cycling, and vegetation growth [2]. Many scholars have conducted extensive research on global warming. Wang et al. [3,4] studied the effects of climate warming on the net primary productivity (NPP) in many ecosystems. Lin et al. [5] studied the effects of grazing and other human activities on the ecological environment

and spring vegetation phenology of the Sanjiangyuan Nature Reserve. Human activities were the main factors that affected vegetation growth in the area 10 years ago [6,7]. However, climate change has become the main factor affecting vegetation degradation recent years [8,9]. Based on the results of the above research, Tan et al. [10,11] also carried out a time series analysis of soil carbon content in the area, and assessed the impact of the soil carbon content on the vegetation phenological phases. In addition, Wang et al. [12,13] studied the change of snow cover in high altitude areas and evaluated the effects on vegetation in the Sanjiangyuan Nature Reserve. In particular, vegetation spring phenology is sensitive to climatic factors, such as temperature and precipitation. Thus, monitoring vegetation spring phenology provides a good reference for the better understanding and prediction of the effects of climate change in this area [14].

Research on vegetation phenology is mainly based on the following two steps: the fitting of the time series of the vegetation index curve and the capturing of the vegetation phenology in the time series curve. Currently, the logistic function, polynomial function [15], Gaussian function and Fourier periodic function fitting methods are frequently used to fit the time-series vegetation index [16]. The vegetation index is symmetrical for a growth and senescence time series, but the fitting effect is poor for mature vegetation when a single function is used [17]. Therefore, the double logistic function and double Gaussian function fitting methods were developed and provide an improved fit.

At present, the threshold method and the maximum in the curvature of time series vegetation index are the remote sensing methods used to monitor the vegetation growing season. The thresholds were defined as NDVI ratios of 20% [8,17–19] (G20) and 50% [20] (G50), which indicate the ratio of the NDVI curve at the specific time. $NDVI_{ratio}$ represents the ratio of the NDVI, and Equation (1) shows how to calculate the $NDVI_{ratio}$.

$$NDVI_{\text{ratio}} = \frac{(NDVI_t - NDVI_{\min})}{(NDVI_{\max} - NDVI_{\min})} \tag{1}$$

where $NDVI_t$ is the NDVI value at a given time t, and $NDVI_{max}$ and $NDVI_{min}$ are, respectively, the maximum and minimum NDVI values in the annual NDVI cycle. Specifically, the green-up date is defined as the first day of the year when $NDVI_{ratio}$ value exceeds 0.2 or 0.5. White [18] and Yu [21] established this method by setting observation points and validated the precision over flat areas. However, the method has relatively low precision over large areas and areas with sparse vegetation [22–24]. Thus, another method involves capturing the curvature of the vegetation index time series curve, which improves universality over high altitude areas and areas with sparse vegetation [15,16].

Many studies have focused on improving the above steps to improve the accuracy of the spring green-up date calculation [23]. However, factors such as the continuity of the vegetation index vary throughout a time series [24]. The vegetation index images during the spring phenological phases are largely dependent on climatic factors [23]. Moreover, the accuracy of determining the vegetation green-up date is affected by the sensor spatial resolution. Thus, totally accurately calculating green-up date is significantly difficult [15]. In addition, the vegetation index derived from the MODIS on the Terra satellite is the most commonly used data type because it has middle spatial and temporal resolution as well as high applicability to surface conditions [1,25]. The monitoring effect differs throughout all types of data sources, and that difference has not been verified [21]. Therefore, to find a suitable product for determining the green-up date, the curve shifts of different MODIS products were compared. CMAPD was proposed to determine green-up date. The coupled logistic and polynomial function was used to fit the time series vegetation index. Specifically, the anomalous points were detected using the nearest neighbor algorithm and replaced with the local threshold determined by GDD and land use type. Furthermore, the ordinary least squares were used to fit a linear regression trend to annual vegetation green-up dates. In addition, the ordinary least squares were also used to analyze the vegetation green-up dates distribution along the elevation gradients.

## 2. Materials and Methods

### 2.1. Study Area

The Sanjiangyuan Nature Reserve is located in the southern part of Qinghai Province, as shown in Figure 1, with an elevation of 3500–5000 m. The Sanjiangyuan Nature Reserve is the hinterland of the Qinghai-Tibet Plateau, which is also referred to as the roof of the world. The Sanjiangyuan Nature Reserve is the source catchment area for major rivers that have nurtured China and the Indo-China peninsula for long periods: the Yangtze River, the Yellow River and the Lancang River. The geographical location is 31°39′ N–36°12′ N and 89°45′ E–102°23′ E, and the administrative area includes 16 counties, including Yushu, Guoluo, Hainan and Huangnan, and Tanggula of Golmud City, with a total area of 305,000 square kilometers. The Sanjiangyuan area is dominated by mountainous landforms [1]. In addition, this region has a typical plateau continental climate and is characterized by alternating hot and cold seasons, distinct wet and dry seasons, small annual temperature differences, large daily temperature differences, long sunshine times, strong radiation, and four seasons [26]. The Sanjiangyuan area is affected by geological movements. There are many alpine mountains and relatively high altitudes, which form an obvious vertical zonal distribution of soil. From high to low elevations, the soil types are alpine cold desert soil, alpine meadow soil, alpine steppe soil, mountain meadow soil, gray cinnamon soil, chestnut soil and mountain forest soil. Among these soil types, alpine meadow soil is the most common type [27]. Swampy meadow soil is also common, and the frozen soil layer exists in high altitude area. Marsh soil, fluvo-aquic soil, peat soil, and aeolian sandy soil are intrazonal soils. The Sanjiangyuan area has a large span, and there are certain differences in climate types. More importantly, the vegetation coverage varies greatly. According to the national land use status classification system, the major land use classes include cultivated land, forestland, grassland, residential sites, industrial, mining land, waters and unused land. The region has large areas of broadleaf forest, coniferous and broadleaf mixed forests, shrubs, meadows, grasslands, swamps and aquatic vegetation, mat vegetation and sparse vegetation. In May 2000, the Sanjiangyuan area was identified as a provincial protected area. In January 2003, the level of the nature reserve was upgraded to the national level [28].

### 2.2. Data and Preprocessing

The data used in this study include remote sensing data, meteorological data, land use data and digital elevation model (DEM) data [29]. The remote sensing data were collected from MODIS and Landsat-8. The meteorological data originated from the national meteorological site data. The land use data were acquired from the Chinese Academy of Sciences data, which were mainly based on Landsat reflectance data. In addition, the DEM data were collected from the Advanced Spaceborne Thermal Emission and Reflection Radiometer (ASTER) instrument and the Shuttle Rader Topography Mission (SRTM).

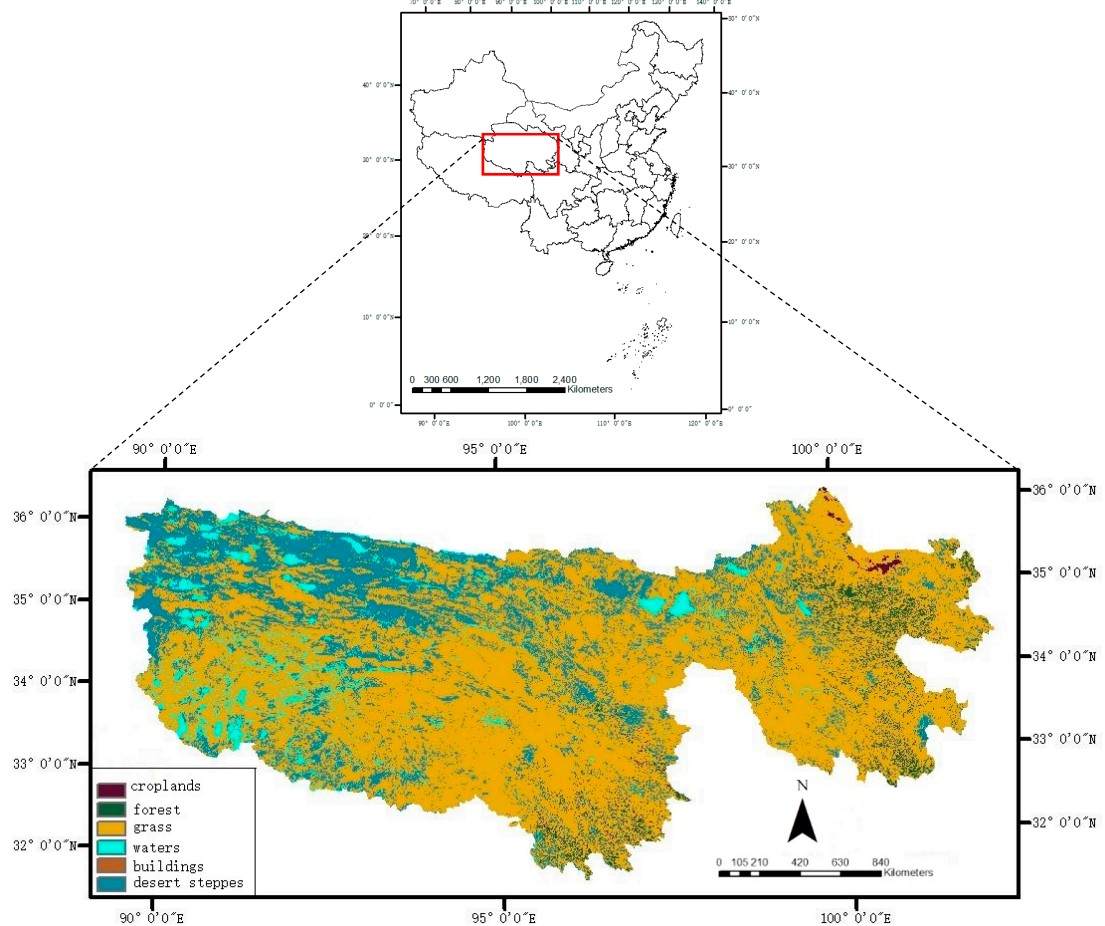

**Figure 1.** The study area.

### 2.2.1. Remote Sensing Data Source and Preprocessing

The remote sensing data include MOD09A1, MOD13Q1, MCD43A4 and Landsat-8 data. The MOD09A1 product is an estimate of surface spectral reflectance. First, the low-level data products are atmospherically corrected and calibrated for aerosols. Then, the products become L2 or L3 basic data products after terrain processing. MOD09A1 provides an 8-day synthetic data product with a range of 1–7 bands in a sinusoidal projection. Each MOD09A1 pixel contains the most likely L2 observations within 8 days, considering the effects of high observation coverage, low viewing angles, cloud shadows and aerosol concentrations. MOD13Q1 products, including NDVI and NOAA Advanced Very High Resolution Radiometer (AVHRR) NDVI products, can be used to demonstrate the shifts of time series vegetation index. The MODIS vegetation index product also includes the enhanced vegetation index (EVI), which is an improved vegetation index using the blue band. MODIS NDVI and EVI products were calculated from two-way atmospherically corrected surface reflectance corrected with water, cloud, heavy aerosol and cloud shadow masks [30]. The vegetation index can be used for monitoring global vegetation growth and can also be used to reflect the surface coverage or the change in surface coverage. The vegetation index can also be used as input data for global biogeochemical and hydrological models, global and regional climate models, biophysical characteristics, terrestrial surfaces and the conversion of terrestrial surface coverage. The MOD43A4 products exhibited improved precision after being adjusted for Bidirectional Reflectance Distribution Function (BRDF) reflectance and the aerosol reflectance was removed from the image. The MOD13Q1 products have a 500 m spatial resolution and 16-day interval. The other MODIS products we obtained have a 500 m spatial resolution and

8-day interval, and the time series is 2000–2016. The data were converted to NDVI and EVI after band calculation, as shown in the following.

$$NDVI = \frac{\rho_{NIR} - \rho_R}{\rho_{NIR} + \rho_R} \tag{2}$$

$$EVI = \frac{2.5(\rho_{NIR} - \rho_R)}{\rho_{NIR} + 6\rho_R - 7.5\rho_B + 1} \tag{3}$$

In these equations, $\rho_{NIR}$ represents the near-infrared band that ranges from 841 to 875 nm; $\rho_R$ represents the red band that ranges from 620 to 670 nm; and $\rho_B$ represents the blue band that ranges from 459 to 479 nm.

The Landsat-8 Operational Land Imager (OLI) is a linear array imaging system, and the satellite can achieve global coverage every 16 days. Landsat-8 is mostly consistent with Landsat-1 to Landsat-7 in terms of spatial resolution. The satellite has 11 bands, and bands 1–7 and 9–11 have a spatial resolution of 30 m. The satellite has panchromatic bands, in which the red band ranges from 630 to 680 nm, the infrared band ranges from 845 to 885 nm, and the blue band ranges from 450 to 550 nm. When MODIS products and Landsat products are compared, the inconsistent wavelength ranges of the bands will cause errors in the results. Therefore, the Landsat reflectance data were processed by the spectral response function derived from the Landsat-8/OLI sensor to reduce the difference caused by the different sensors [31]. Because of the high resolution of Landsat, it reflects the vegetation index curve precisely for a specific type. We used Landsat data to choose the best product for green-up date calculation.

### 2.2.2. Other Database and Processing

The meteorological data are from the meteorological sites that the National Weather Service deployed throughout the country to meet the needs of national meteorological services. The meteorological data include temperature (°C), precipitation (mm), wind speed (m/s) and other factors. The acquisition of mean temperature data was 1 day from 20 meteorological sites in the Sanjiangyuan area [32], and the time series is 2000–2016. These data were used for eliminating anomalous points in the green-up date calculation method.

The 30 m resolution land use classification map was rasterized by the 1:100,000 land use vector map (obtained by Landsat image visual interpretation) provided by the Institute of Remote Sensing and Digital Earth of the Chinese Academy of Sciences. Initially, the 30 m land use classification map was used to monitor China's ecological environment and proved to be highly precise. These data were used to choose the pure pixels (480 m pixel contains the same land use type) from MODIS for determining the best data for green-up date calculation. Then, to match the green-up date images, the 30 m resolution land use classification map was also resampled to 500 m. Because it is difficult to directly resample 30 m images to 480 m. First, the 30 m images were resampled to 480 m (256 pixels of 30 m). The value, which filled the most pixels in the 278 pixels of 30 m, and assigned it to the 480 m pixel. Then, we resampled from 480 m to 500 m using nearest neighbor method. The main role of the 500 m land use map was to determine the vegetation type and eliminate the anomalous points of the green-up date images. In fact, high-accuracy land classification maps obtained by other methods can replace the land use data used in this study [33].

This study also used elevation data from ASTER DEM version 1 (A1), which is a digital elevation data product with a global spatial resolution of 30 m. After splicing and resampling, the product became 500 m elevation data, which were used to analyze the green-up dates distribution along the elevation gradients.

### 2.3. CMAPD Method

　　Different from the existing green-up date calculation method, the CMAPD method optimized the processes of green-up date calculation. First, we chose the best MODIS vegetation product for green-up date calculation and extracted the time series vegetation index from remote sensing images. Second, we used the coupling of logistic and polynomial functions to fit the time series vegetation index. Third, the initial green-up date corresponded to the times at which the curvature in the vegetation index exhibited local maximums. Fourth, the anomalous points of green-up date were detected using the nearest neighbor algorithm. Finally, the anomalous points were replaced by local threshold determined using GDD and land use type. The overall flowchart is shown in Figure 2. The calculation method of the key growth period of vegetation is detailed in Sections 2.3.1–2.3.5.

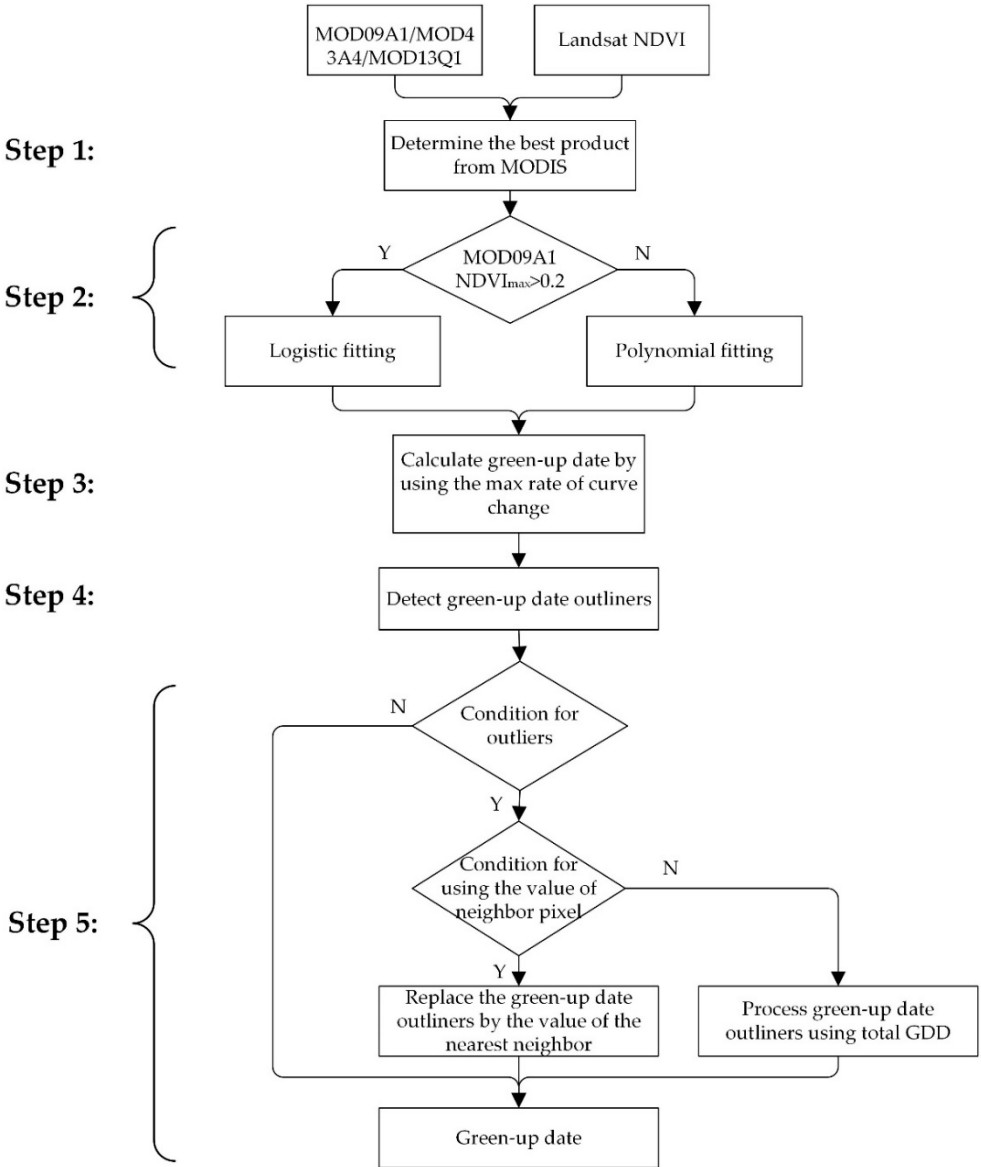

**Figure 2.** Method flowchart.

### 2.3.1. Step 1: Determining the Best NDVI Time Series

　　To choose the best data for calculating the green-up date in the Sanjiangyuan Nature Reserve, we compared the shifts of curve from different MODIS products, including NDVI and EVI calculated by

MOD43A1 products, NDVI and EVI with MOD09A1 reflectance products, and MOD13Q1 products. The time series vegetation index was extracted from above products.

However, the spatial resolutions of Landsat data and MODIS data are different. It is not reasonable to directly compare the change point for time-series vegetation index. However, when the Landsat and MODIS pixels in the same location contain the same pure type of vegetation, the two products can be compared. To find the pixels with the same vegetation type (pure pixels) and match the 30 m Landsat remote sensing images, all MODIS images were resampled to 480 m by the nearest neighbor algorithm [29]. A MODIS pixel theoretically contains 256 land use pixels, which has 30 m resolution. The big pixel was defined as pure when more than 95% of the 30 m pixels in a window belonged to the same land use type. The cell was marked as a specific endmember. Therefore, the time series curve of the MODIS vegetation index was obtained from pure pixels. According to this method, we obtained time series vegetation index for different vegetation types. Finally, the curve from different MODIS products were compared with the vegetation index from Landsat. Specifically, the product most consistent with the trend of the Landsat NDVI curve was used for green-up date calculation.

### 2.3.2. Step 2: Fitting Time-Series Curve of the Vegetation Index Using a Coupled Model

Remote sensing images are used to capture vegetation shifts in the process of Earth observation. The scatter plot comprising remote sensing images at a certain time can almost reflect the growth and changes in vegetation. In this study, the MODIS remote sensing image map of the Sanjiangyuan Nature Reserve for each year were used to extract the NDVI pixel by pixel every 8 days. Each pixel can experience a complete cycle of vegetation growth every year from 2001 to 2016. Although the time series curve of the vegetation index is not an ideal curve, it can be fitted with relatively high precision using the logistic function [33,34]. However, the maximum value of the vegetation index is relatively small in one year for sparse vegetation. The altitudinal gradient in the Sanjiangyuan Nature Reserve is large, and the highest altitude reaches 6000 m. Therefore, the vegetation index is easily influenced by extreme climatic factors, such as snowfall. The growth of vegetation will be seriously disturbed under these conditions. Therefore, this study used a coupled model including logistic and polynomial functions to fit the time series NDVI [15].

$$NDVI(t) = \begin{cases} d + \frac{c}{1+\exp(a+bt)}, NDVI_{max} > 0.2 \\ mt^5 + nt^4 + gt^3 + ht^2 + kt + f, NDVI_{max} \le 0.2 \end{cases} \tag{4}$$

where $t$ is the day of year. $NDVI(t)$ is the NDVI on day $t$ of the year, and $a$ and $b$ are the fitting parameters. $c$ plus $d$ is the maximum NDVI and $d$ is the NDVI of the initial background field value. Additionally, $m$, $n$, $g$, $h$, $k$ and $f$ are the fitting parameters for the polynomial function. After comparing the root mean square error (RMSE) of different thresholds for the coupled models, when $VI_{max} > 0.2$, the logistic model was used to fit the time series of the vegetation index, and when $VI_{max} \le 0.2$, the polynomial fitting method was adopted. The fitting error is shown as follows.

$$RMSE = \sqrt{\frac{(NDVI(t) - M(t))^2}{N}} \tag{5}$$

The number of days of MODIS observations is denoted $t$, $NDVI(t)$ is the NDVI value of the fitted curve in days $t$, $M(t)$ is the MODIS NDVI observations, and $N$ is the number of observations of the NDVI in the year.

### 2.3.3. Step 3: Calculating Green-Up Date by Maximum Curvature

To calculate green-up date, maximum curvature in the fitted curve was captured under the condition that green-up date is distributed within day 50–180 of year according to the local climate

and vegetation phenology observations. Under the above condition, the fitting curve of time series vegetation index is subjected to secondary derivation as follows.

$$\partial'' = \begin{cases} \frac{b^3cz(3z(1-z)(1+z)^3(2(1+z)^3+b^2c^2z))}{((1+z)^4+(bcz^2)^{\frac{5}{2}}} + \frac{b^3cz(1+z)^2(1+2z-5z^2)}{((1+z)^4+(bcz^2)^{\frac{3}{2}}}, NDVI_{max} > 0.2 \\ 20mt^3 + 12nt^2 + 6gt + 2h, NDVI_{max} \leq 0.2 \end{cases} \tag{6}$$

where $z = e^{a+bt}$. During the growth period, when $\partial''$ reaches the maximum under the condition, the initial green-up date can be inferred from Equation (6).

2.3.4. Step 4: Detecting the Anomalous Points

The time-series curve of the remote sensing vegetation index is affected by many factors, such as clouds and water vapor. It also reflects the anomalousness of vegetation growth and shifts. Thus, vegetation will be affected by human activities and extreme climatic conditions during the vegetation growing season. The interference of these external factors will also lead to anomalous vegetation growth, and the detection of vegetation phenology is one of the ways to monitor vegetation change. However, the phenological phases of vegetation are difficult to capture for some extent due to the poor growth of vegetation or sudden disasters. Therefore, the existing methods for calculating green-up date are devoted to improving the time series vegetation index fitting method and the seasonal shifts detection method. There are few studies on the regional spring green-up date. Based on the initial green-up date image through Step 3, this study also introduced a $3 \times 3$ sliding window, in which the center pixel in the window is labeled N, and the surrounding 8 pixels are N1, N2, N3, N4, N5, N6, N7 and N8. Each pixel corresponds to the onset of vegetation green-up, and the surrounding pixels are used to determine if the central pixel is an anomalous point.

$$S = \sqrt{\frac{(GUD_N - GUD)^2 + (GUD_N - GUD_{N2})^2 + \cdots + (GUD_N - GUD_{N8})^2}{8}} \tag{7}$$

where $S$ is the difference between the green-up date of the central pixel and that of the surrounding pixels. $GUD_{NX}$ represents the onset date of vegetation green-up corresponding to each pixel in the sliding window. However, 70% pixels of $S$ in this region is less than 28. When $S$ is larger than 28, the central pixel is considered anomalous in this study.

2.3.5. Step 5: Replacing Anomalous Points Using the Local Threshold

Anomalous points detected in Step 4 are flagged, and these cells were classified according to the land use type. In reality, the spring green-up date is sensitive to the characterization of species and climate factors [35]. Therefore, the threshold method was used to correct the anomalous points. If the land use type of the anomalous point is the same as the surrounding vegetation, then we used the nearest pixel values [36] to fill the anomalous points.

In other cases, the anomalous points were corrected in combination with the temperature. Wang et al. [37–39] proposed that vegetation phenology is sensitive to temperature during vegetation growing season. Specifically, vegetation growth is strongly correlated with growing degree-days (GDD) [40,41]. The formula for calculating annual total GDD (in °C·day) is shown as Equation (8).

$$GDD = \sum_{t1}^{t2} \left[ \left( \frac{T_{MAX} + T_{MIN}}{2} \right) - T_{base} \right] \tag{8}$$

where t1 represents day of year when the average day temperature is below 0 °C and t2 represent the onset date of vegetation green-up. $T_{MAX}$ (in °C) and $T_{MIN}$ (in °C) represent the ground maximum and minimum temperature of a certain day, respectively. $T_{base}$ (in °C) is the base temperature below which vegetation ceases to be biologically active. $T_{base}$ varies among species and possibly cultivars,

and likely varies with growth stage or process being considered [42]. The $T_{base}$ typically used is 0 °C in the Sanjiangyuan Nature Reserve [43]. Based on this, we used total GDD for each vegetation type as an important indicator for green-up date calculation [44].

The Sanjiangyuan area is vast, and there are only 20 meteorological stations in this area. The total GDD was calculated within 10 km of each meteorological station in the Sanjiangyuan area (more than 100 pixels for each land use type). Then, the average GDD was calculated from 2000 to 2016. The total GDD of spring phenological phases and standard deviation of each land use type are shown in Table 1. The standard deviation of total GDD is between 1 and 2.5 °C, indicating that there is a strong correlation between total GDD and green-up date. Therefore, the date at which the total GDD reaches the GDD for different vegetation type mentioned in Table 1 was considered to be the green-up date for the pixel. This method was used to replace the anomalous points in the green-up date images.

**Table 1.** The total GDD before green-up date for each vegetation type.

| Vegetation Type | GDD (Unit: °C) | Standard Deviation |
|---|---|---|
| Grass | 8.17 | 1.38 |
| Cropland | 8.56 | 2.45 |
| Forest | 9.29 | 1.65 |

*2.4. Analyses*

The ordinary least squares and Mann–Kendall tests [45] were used to fit a linear regression trend to annual vegetation green-up dates from 2000 to 2016. We only considered the vegetation land use type, which is available during the study period. Then, we analyzed the relationship between vegetation green-up dates and altitudes. The ordinary least squares were used to calculated the slope and distribution along the altitude gradient for every 100 m in a sub region (more than 500 pixels).

**3. Results**

CMAPD was used to calculate the 2016 spring green-up date in the Sanjiangyuan Nature Reserve, as shown in Figure 3a. The green-up date in the east is obviously later than that in the west, which is related to the altitude situation, as the east of the Sanjiangyuan Nature Reserve is lower than the west. In addition, the trend of the green-up date from north to south is gradually advanced, which is related to the latitude, which is consistent with the research results of Zhang et al. [16]. However, the vegetation is sparse in the central part of the Sanjiangyuan Nature Reserve, and the green-up date is later than other areas, which is related to the high altitude. Obviously, the green-up date calculated by CMAPD is spatially continuous, which is consistent with the continuous distribution of vegetation.

*3.1. Comparison of the Results for Whole Study Area*

To analyze the spring green-up date obtained by CMAPD (Figure 3a), we compared it with Zhang's green-up date calculation method [16]. This method uses a piecewise logistic model (PL) to fit time series vegetation indices, which has been adopted to produce the global phenology product (MOD12Q2). To match the results of the anomalous points correction, Zhang's results were smoothed using the nearest neighbor, as shown in Figure 3b. The absolute value of the difference between the two products of spring green-up date (Bias) is shown in Figure 3c.

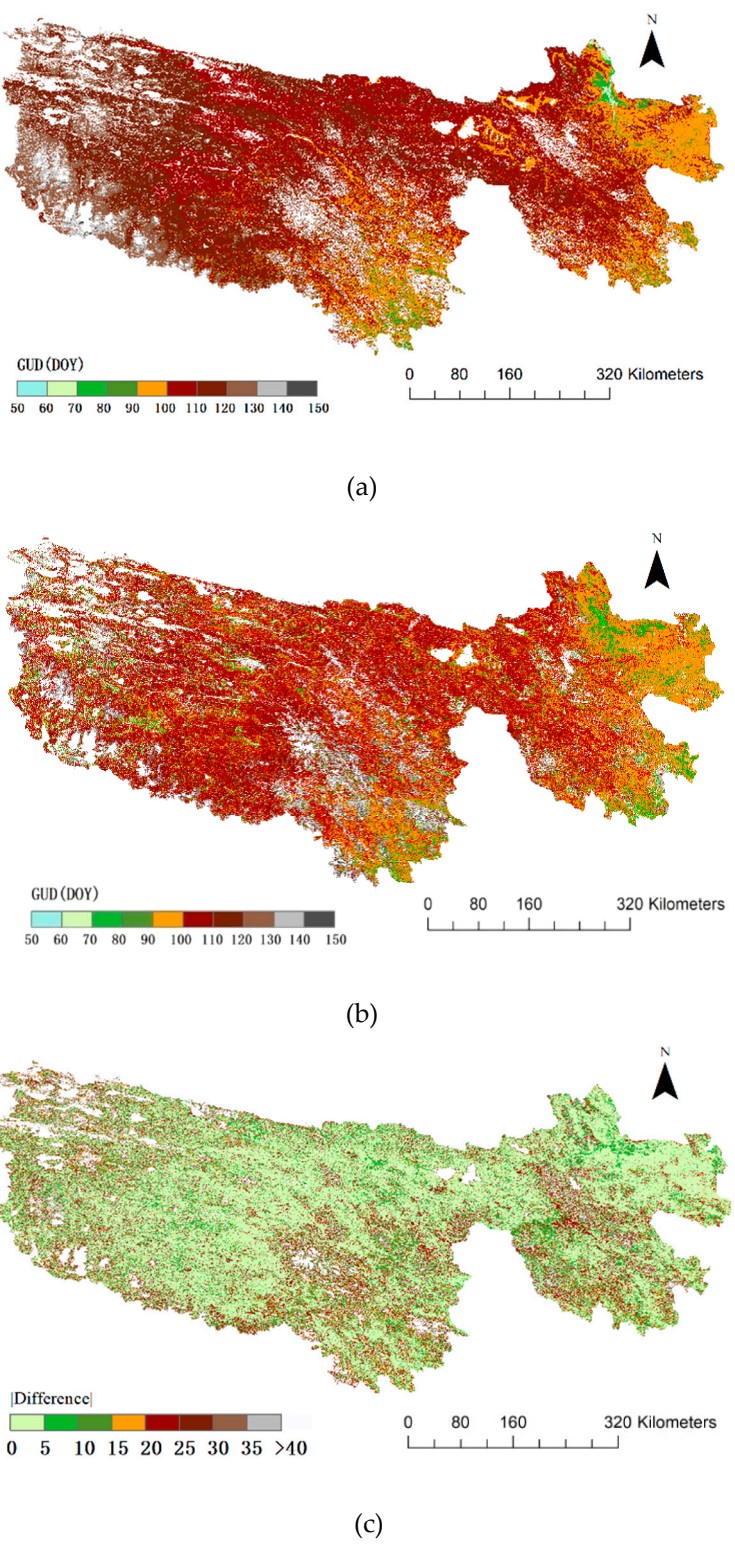

**Figure 3.** Comparison of the results of the calculation methods for the vegetation greening period in 2016: (**a**) image of green-up date using the CMAPD method; (**b**) image of green-up date from Zhang's methods; and (**c**) the absolute value of the difference of the two spring green-up date products (bias).

The onset date of vegetation green-up is not only affected by the composition of the vegetation in the image but also the climate, especially spring temperature. In ecology, the growth and development of vegetation and the spring temperature show a strong correlation [40]. Due to the special topography

and climatic factors in the Sanjiangyuan Nature Reserve, the vegetation growth is strongly affected by altitude and sudden natural disasters. Therefore, the results of the calculation of the green-up date using CMAPD are shown in Figure 3a. As shown in Figure 3, the green-up date in the western and southern areas of the Sanjiangyuan Nature Reserve is significantly later than the green-up date product from Zhang's method. However, in the northeastern part of the Sanjiangyuan Nature Reserve, the green-up date is significantly earlier. In the west of the study area, the elevation is more than 5000 meters above sea level, the coverage of vegetation is sparse, and it is greatly affected by sudden snowfall. The CMAPD is more suitable for areas with sparse vegetation. In the absence of processing anomalous points, there are many anomalous points in the area. The green-up date calculated by CMAPD shows good consistency, and the overall effect is consistent with the topography in the Sanjiangyuan Nature Reserve.

### 3.2. Comparison of the Results in A Sub Region

Over a large area, the green-up date calculation method exhibits improved regional adaptability. To observe the performance of CMAPD at detecting anomalous points, a sub region $(120 \times 120 pixels)$ as selected, as shown in Figure 4.

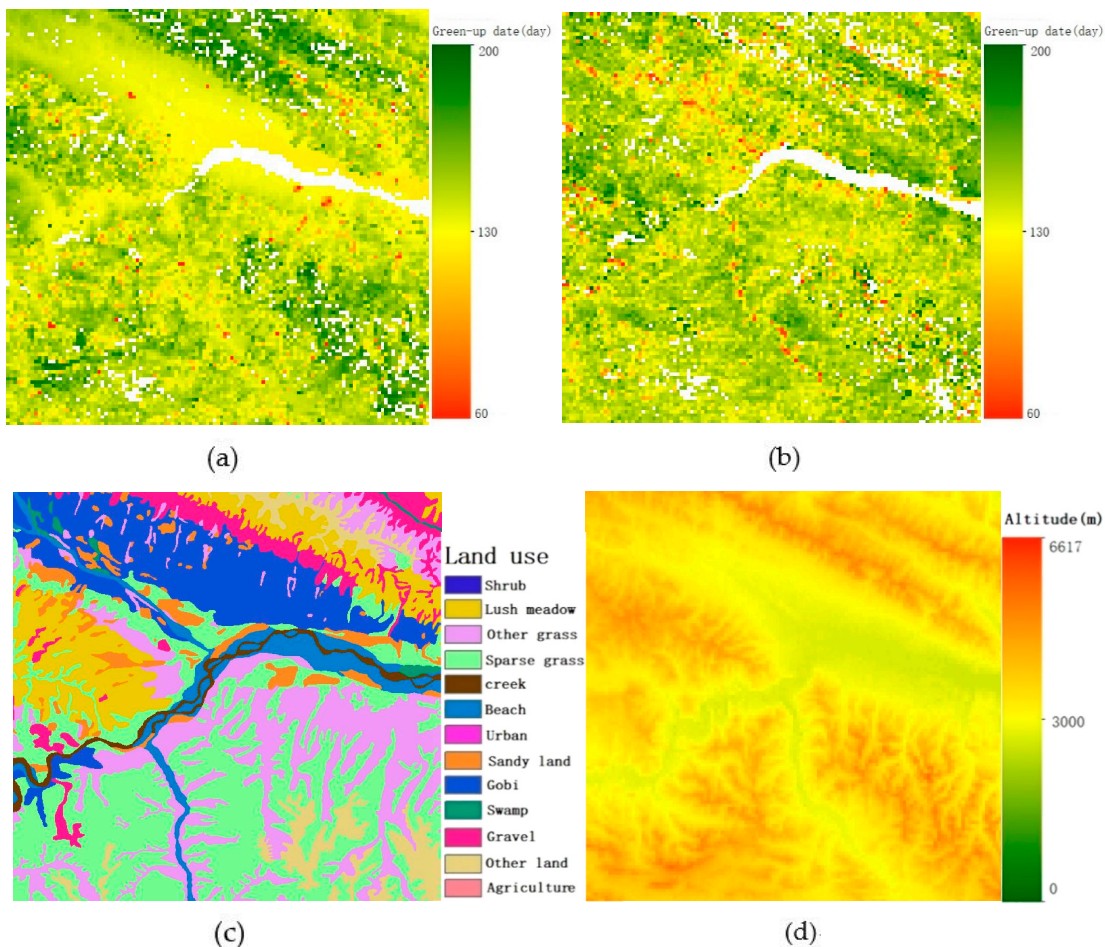

**Figure 4.** Comparison of the local green-up date and influence factors analysis: (**a**) green-up date using the CMAPD method; (**b**) green-up date product from Zhang's method; (**c**) local area land use type; and (**d**) DEM of local area.

As shown in Figure 4a,b, the green-up date exhibits a similar spatial distribution. However, the spatial distribution of green-up date from CMAPD has a stronger continuity than that from Zhang's method. According to statistical analysis, about 36.7% of the selected area (Figure 4) is anomalous. The

main land use types of anomalous points are shrubs and grass. In addition, about 29% of the whole selected area was replaced. This reflects that most of the anomalous points were eliminated when using the CMAPD method. Figure 4c is a land use type map, which shows that the green-up date obtained by the CMAPD exhibits strong agreement with the land use type in the local area. In addition, the green-up date calculated by CMAPD shows a good agreement with the DEM image shown in Figure 4d. Similar to other studies [46,47], the onset date of green-up is significantly influenced by temperature and DEM. According to the analysis in sub region, the green-up date calculated by CMAPD is strongly related to land use type and altitude. In summary, the green-up date calculated by CMAPD has continuity and consistency in spatial distribution.

The NDVI time series were extracted from the selected sub region (Figure 4) by extracting the vegetation types, and the average value was calculated for each Julian day. The time series formed by these average values reflect the vegetation growth in this region, as shown in Figure 5.

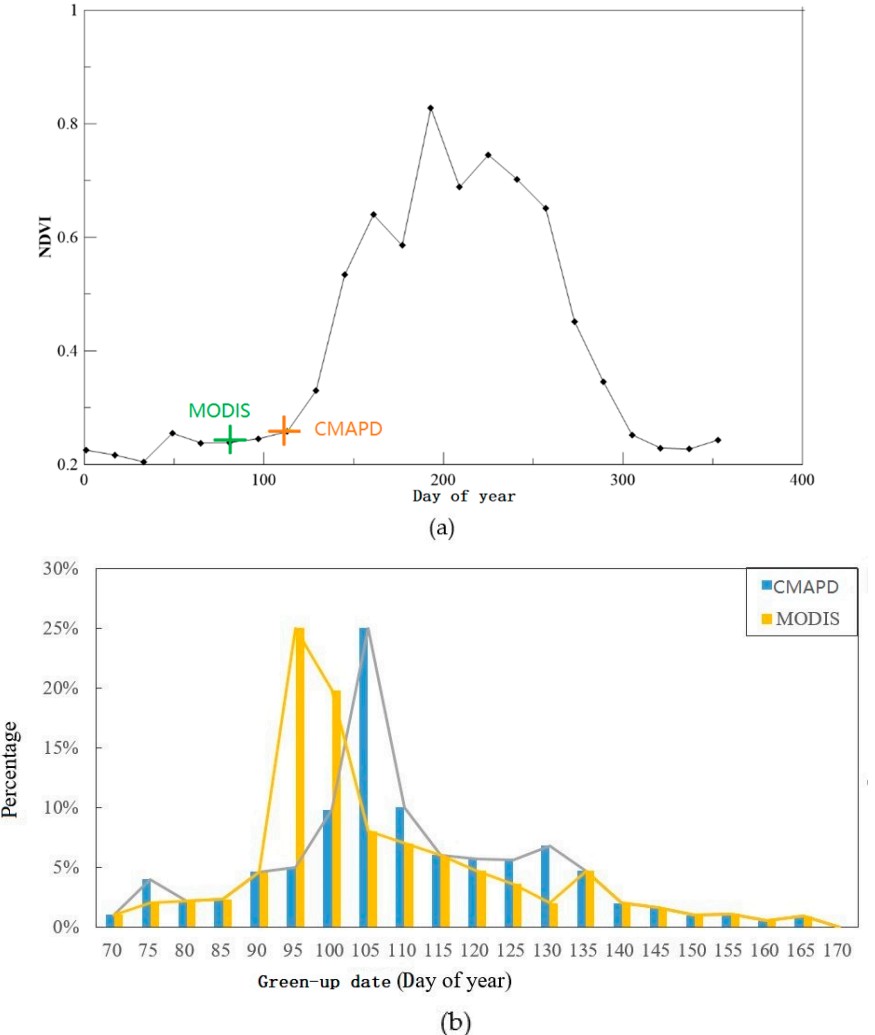

**Figure 5.** Green-up date pixel statistic: (**a**) regional green-up date frequency map; and (**b**) regional time series NDVI change diagram.

In Figure 5a, the onset date of the vegetation green-up when the vegetation began to change was Day 105, while the average green-up date from Zhang's result was Day 95. Figure 5b is a frequency diagram of the green-up date. It can be seen that the value calculated by the CMAPD is mostly distributed on the Day 105, while the green-up date images from Zhang's result is mostly distributed on Day 95. However, the point at which the time series NDVI curve begins to change is roughly Day

105. This is consistent with the average green-up date calculated using CMAPD. This further confirms that it is reasonable to use CMAPD to determine the spring green-up date, and the results are consistent with the actual spring green-up date. Therefore, this method effectively eliminates anomalous points.

### 3.3. Change in Vegetation Green-Up Date from CMAPD

#### 3.3.1. Temporal Trends of Green-Up Date at the Regional Scale

The spring phenological phases of vegetation are affected by the land use type and climatic factors. In the study area, because of the establishment of nature reserves, the land use type has not changed greatly since 2000. Therefore, it can be inferred that climatic factors are the main factors leading to changes in vegetation during the spring phenological phases. Therefore, accurate monitoring of the trend and distribution of green-up date is of great significance for monitoring the changes in vegetation and even monitoring the climate changes. This study used CMAPD to calculate annual vegetation green-up date from 2000 to 2016. The ordinary least square was used to fit a linear regression trend to annual vegetation green-up date, which was the average for the whole study area.

Figure 6 illustrates the interannual variations of the vegetation green-up date from 2000 to 2016. The green-up date during 2000–2016 showed an advance trend, but this trend was not significant, similar to the previous study of Piao [48]. Figure 6 shows the trend of the average green-up date for all pixels in the Sanjiangyuan area, which was not significant. To analyze the spatial distribution of the change trend in the Sanjiangyuan area, the Mann–Kendall method [45] was used to calculate the trend pixel-by-pixel.

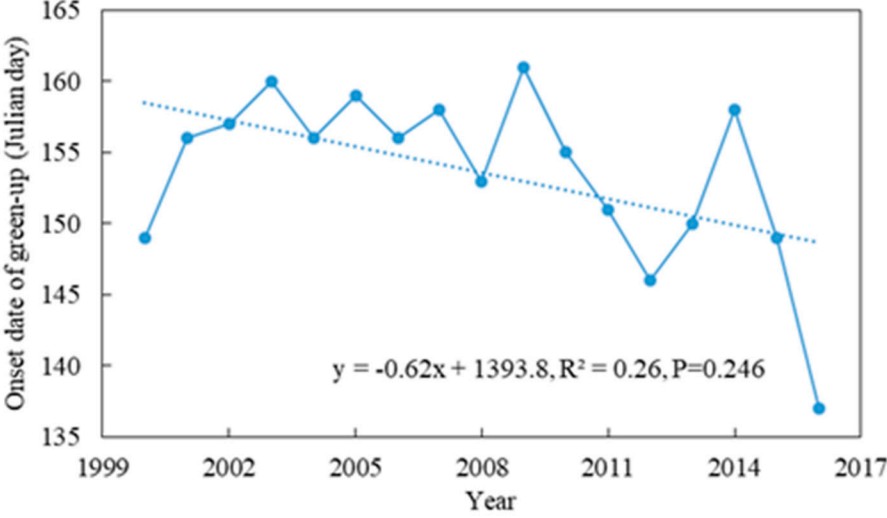

**Figure 6.** Interannual variation of green-up date.

The spatial patterns of trends in green-up date are shown in Figure 7 from 2000 to 2016. The trend of average vegetation green-up date is not significant (Figure 6). In addition, the vegetation green-up date in most areas of the Sanjiangyuan showed an insignificant advance trend according to Figure 7. However, there are still some areas where the vegetation green-up date shows a significant change trend. According to statistical analysis, 10.4% of the region showed a significant advance trend, and 2.2% of the regions showed a significant delay trend. These pixels are mainly distributed in the eastern and western areas of the Sanjiangyuan area, as well as the central areas.

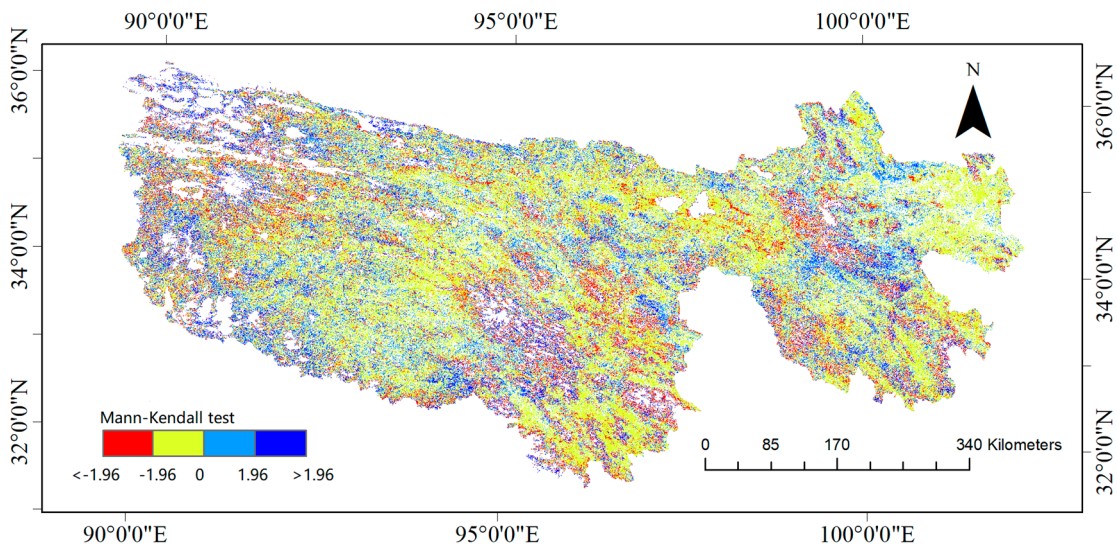

**Figure 7.** Spatial distribution of Mann–Kendall test in green-up date from 2000 to 2016.

### 3.3.2. Vegetation Green-Up Date in Relation to Elevation

To analyze the distribution of vegetation green-up date in different altitudes, a region with more than 500 pixels was selected. The altitudes in this region are distributed within 3500–5500 m. The least squares method was used to fit a linear regression trend of green-up date to every 200 m altitude gradients. The results for 2004, 2008, 2012 and 2016 are shown in Figure 8.

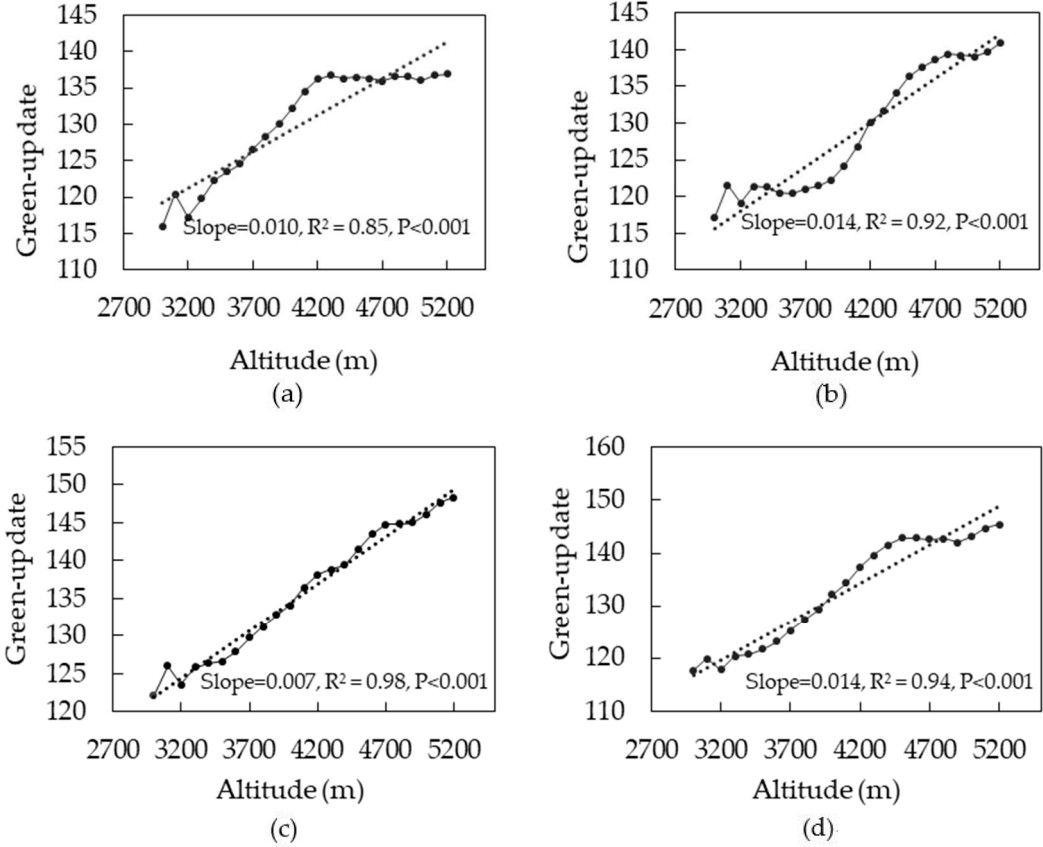

**Figure 8.** Vegetation green-up date along altitudinal gradient in a sub region: (**a**) green-up date along altitudinal gradient in 2004; (**b**) green-up date along altitudinal gradient in 2008; (**c**) green-up date along altitudinal gradient in 2012; and (**d**) green-up date along altitudinal gradient in 2016.

The distribution of vegetation green-up date is significantly related to altitude (Figure 8a–d, *P* < 0.001). Spring vegetation turns green earlier in lower elevations, which is in line with previous findings [48,49]. Oppositely, it was observed that the vegetation green-up date is delayed as the altitude increases ($R2 > 0.85$, $P < 0.001$) at a rate of 0.7–1.4 days per 100 m ($P < 0.001$). Such delayed green-up was more obvious in 2008 and 2016 (1.4 days per 100 m). However, the green-up delay was smaller in 2012 (0.7 days per 100 m). The average rate for 16 years was larger than the rate found by Piao in Qinghai-Xizang Plateau (0.8 days per 100 m) [48].

## 4. Discussion

### 4.1. Determination of the Fitting Function and Its Threshold

The calculation of spring green-up date using CMAPD has some obvious advantages. First, vegetation time series curves can be well fitted in regions with sparse vegetation. For fragile areas, the land use type changes a lot from sparse vegetation to soil within a year (grass to sand, grass to snow, etc.), and the Sanjiangyuan Nature Reserve, which has a high altitude and cold climate, is covered by snow during most of the year. Therefore, the growth of vegetation is highly susceptible to interference. In the CMAPD method, the accuracy of fitting function is very important for the determination of green-up date. The threshold for the coupled model is the key step for determining green-up date.

In this study, 20 pixels were randomly extracted in the selected area (Figure 4a). The time series NDVI for grass was used to test the accuracy of different fitting functions and thresholds. We compared Logistic function, polynomial function and coupled models, respectively, and calculated the fitting error (RMSE) of the three functions. The thresholds and RMSEs are shown in Table 2.

**Table 2.** RMSE of three fitting functions when taking different thresholds.

| Threshold of NDVI | RMSE | | |
|:---:|:---:|:---:|:---:|
| | **Logistic Function** | **Polynomial Function** | **Coupled Model** |
| 0.10 | 0.276 | 0.280 | 0.277 |
| 0.14 | 0.235 | 0.364 | 0.456 |
| 0.18 | 0.260 | 0.381 | 0.321 |
| *0.20* | 0.195 | 0.316 | ***0.173*** |
| 0.24 | 0.243 | 0.314 | 0.198 |
| 0.28 | 0.314 | 0.245 | 0.287 |
| 0.30 | 0.345 | 0.348 | 0.257 |
| 0.34 | 0.310 | 0.262 | 0.269 |

When the maximum value of the vegetation index in a pixel was greater than 0.2, the logistic function was used to fit the time series curve. When the maximum value was less than 0.2, the polynomial function was used to fit the time series curve. In other words, the fitting accuracy of the time series NDVI and the threshold of the curvature largely depend on the maximum value of the vegetation index.

### 4.2. Data selection for Calculating Spring Green-Up Date

The spring phenological phases is affected by other factors. Because the synthesis algorithms for MODIS vegetation index products from different sources differ, there are some differences in green-up date calculation using different reflectivities. The pure pixels of grassland in the Sanjiangyuan area were selected (more than 100 pixels), the average vegetation indices corresponding to the pixel were calculated from different products, and the change points were compared, as shown in Figure 9.

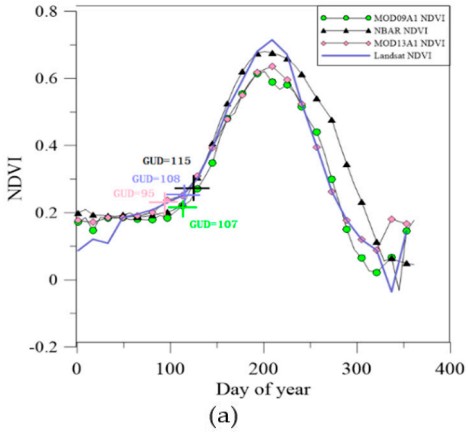 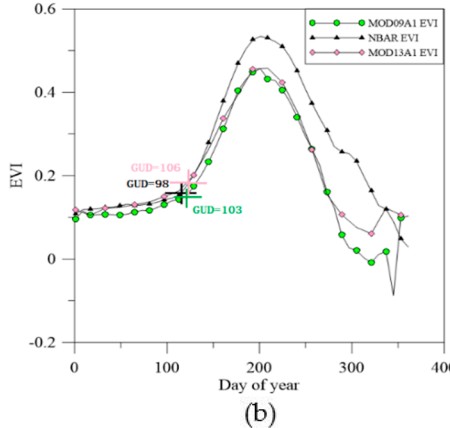

(a)  (b)

**Figure 9.** Comparison of trends in different MODIS vegetation index products: (**a**) four sources of NDVI; and (**b**) three sources of EVI.

In Figure 9, there are several rates of change: first, the vegetation shifts detected by EVI is always approximately 2–5 days earlier than that detected by NDVI. The main reason is that EVI considers the soil background, and the detection of vegetation is sensitive; it is related to data system errors. Second, when using MOD09A1 data to detect vegetation shifts, it is always approximately one week later than that detected by the other two synthetic data. The reason may be that the Nadir BRDF-Adjusted Reflectance (NBAR) data are different from the algorithm used by MOD13A1, and that difference is related to a data system error. Figure 9 compares the use of Landsat with MODIS pure pixels, which indicates that the vegetation spring phenological phases calculated by Landsat reflectivity is consistent with the MOD09A1 data. Therefore, in the subsequent shift detection tests, MOD09A1 quality control data were used to synthesize NDVI to calculate the spring green-up date.

*4.3. Insufficiencies and Prospects for CMAPD*

The research in this paper was based on the idea that the spring green-up date in the region is mainly affected by vegetation type, local climatic factors and elevation [50]. However, in reality, the vegetation phenology is also affected by a series of factors such as precipitation, soil and the ecosystem carbon cycle. Therefore, there is no perfect model to accurately calculate the green-up date. In addition, the growth of vegetation also exhibits certain contingency. The use of remote sensing images provides a rough estimate of the regional spring green-up date. In addition, the estimation of the green-up date can provide an important reference for regional climate change, while the long-term vegetation patterns are affected by both vegetation type changes and climatic factors; therefore, it is necessary but very difficult to thoroughly understand the phenological phases of vegetation. The method used to fit the vegetation index curve and monitor the phenological phases of vegetation through the characteristics of the curve is an important vegetation phenology monitoring method [35], which opens the opportunity for remote sensing monitoring of vegetation phenology. This method is, to some extent, a type of data modeling method. Based on this, the study considered the factors the influence phenology, but the factors considered are far from enough. A follow-up study can consider more phenological events, which would be a direction for further research.

Gathering knowledge of the quality of a model, in addition to its sensitivity, robustness, applicability, and the accuracy of its results, is the first step of model building. The research conducted by remote sensing satellites reflects the overall shifts in large areas. However, due to the limitation of satellite sensors and the influence of spectral transmission media, remote sensing images often have certain errors when reflecting the surface conditions. Therefore, when remote sensing satellites are used to monitor surface shifts, they need to be combined with surface measured data. In the case of mutual verification, the accuracy of remote sensing data can be guaranteed. However, there are many uncertain factors in the measurement of vegetation phenology. For example, the onset

dates of green-up for different vegetation types are not uniform. Different observation methods have different judgments on the vegetation phenology. The phenological shifts in vegetation are difficult to capture. The extremely limited range and other factors results in greater unreliability of the vegetation phenological field observation data. Therefore, this study used the comparison method and prediction method to verify the CMAPD method. The above method can rationally be used to verify the results, but it is not accurate. The development of perfect vegetation phenology observation indicators is the key factor to ensure the availability of vegetation phenological field observation data. Furthermore, phenological cameras [17] have gradually become supplementary data that can be used to verify vegetation phenology field observation data.

## 5. Conclusions

In this study, we proposed an improved spring green-up date model called CMAPD. We describe the research basis, application, and performance of the method as used in the Sanjiangyuan Nature Reserve. The time series vegetation index is fitted using the logistic and polynomial fitting methods, and then the curve of the fitting is accurately captured. The anomalous points are detected by neighbor pixels. Finally, the local threshold is used to replace the anomalous points in the spring green-up date images. Compared with the spring green-up date product from Zhang's method, this model has the following two advantages according to the experiment. First, the model has a better fitting effect on the vegetation variation curve for sparse vegetation, and the obtained spring green-up date is more accurate. Especially for high-altitude areas, the disadvantages of the calculation of the onset date of vegetation green-up due to the sparse vegetation are effectively avoided. Second, the model can effectively correct the regional anomalous points by using the local threshold. Given these advantages, we conclude that the CMAPD can obtain better results for the regions of sparse vegetation. In addition, a weak statistically significant advance trend for average vegetation green-up date was observed from 2000 to 2016. However, 10.4% of the study area has a significant advance trend. Regression analysis shows that the green-up date is correlated to elevation: the green-up date is clearly later in higher elevations. Moreover, the CMAPD provides new ideas and inspiration for calculating vegetation phenology and analyzing the trend of vegetation phenology.

**Author Contributions:** J.S. conceived and designed the study. Q.L. contributed to the conception of the study, performed the data analysis and wrote the paper. J.S. and L.Y. reviewed and edited the manuscript. Methodology, J.W.

**Acknowledgments:** This research was supported in part by the National Key Research and Development Program of China (2016YFB0501502) and the National Natural Science Foundation of China (No. 41871231).

**Conflicts of Interest:** The authors declare no conflicts of interest.

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
