# Peer review of "Improved Spring Vegetation Phenology Calculation Method Using a Coupled Model and Anomalous Point Detection"

_remotesensing, doi:10.3390/rs11121432_

Round 1

Reviewer 1 Report

General Comments on Remote Sensing manuscript 494934

This paper is on an interesting topic and presents some methods that have promise for green-up analysis. I have made numerous minor comments in the manuscript pdf and will summarize my main comments here.

1. This paper is about methods, but important details of the methods testing are not reported. For example, the paper uses a polynomial regression when the maximum NDVI is less than 0.2 and a logistic regression when it is higher than 0.2. The testing that led to this choice is not reported and the choice is not justified. Anomalous pixels are identified as having a S>28 (S as defined in equation 8); the choice of this value is not justified. The values for growing degree-day thresholds used to estimate green-up (Table 1) are simply presented without any source or justification.

2. The main comparision is phrased as being between CMAPD and "MODIS". In fact CMAPD is based on MODIS data too, and I was not able to determine exactly how the inferior "MODIS" method differed from the preferred CMAPD method. The details are probably in the Zhang et al 2003 paper; if they are going to use Zhang's method as the sole standard to judge their method against, they need to explain what it is and how they differ.

3. On line 149 the paper states that the data were all resampled to 16-day composites. I'm not sure exactly how this could be done, but in any case I would never want to degrade the temporal resolution of the 8-day data. In alpine and arid regions green-up can be very fast, and I would say that 16-day composites are too coarse to locate a green-up date.

4. Several plots of NDVI or EVI vs. day of the year are presented, but we never learn what the source of the NDVI (EVI) values is: for example, is it an average of the study area, or part of the study area?

5. The temperature maps used to replace anomalous values: the paper states that these were obtained by kriging, with no information about the weather stations used as input to the kriging. I assume that the weather stations are sparse, irregularly distributed, and mostly in valley locations where the settlements are. I don't see how kriging could give you accurate daily temperature estimates for a huge mountainous area like this, where elevations vary greatly from the weather stations. It seems to me that some kind of crude model that accounts for elevation (using a lapse rate) would be needed at the least. If the authors want to use simple kriging then they need to supply some information to show that this method works here.

6. The Discussion mentions an interest in trends with time or the effects of environmental factors on green-up dates. These kinds of analyses would make the paper more interesting. If the authors want to keep it as strictly a methods paper, then better quantitative analysis of the accuracy of the methods are needed. At the least this would involve quantifying where and how much the CMAPD methods differs and what reasons we have to believe that it is better.

It appears that the authors have done a lot of work to refine their method and have an innovation that could be useful to others in the future. However, to convince the readers I think they need to provide better support for their proposed method.

Author Response

Dear Reviewer,

We thank your comments and suggestions that are very helpful for improving our manuscript (remotesensing-494934). We appreciate your responsible attitude toward our paper, as well as your impressive comments. Your opinion has a great promotion for our research. There is a significant improvement for our study on vegetation phenology, as well as writing skills. Thank you again. We have done our best to revise the manuscript according to your comments. Our item-by-item response to your comments is in red fonts in the PDF file. The major modifications of the paper are marked by yellow background.

Yours sincerely,

Reviewer 2 Report

Article Name:  Improved Spring Vegetation Phenology Calculation Method using a Coupled Model and Anomalous Point Detection

L73: the equations must be separated from the main text, and placed as figure.

L74: “exceeds 0.2 and 0.5” needs explanation.

L78: “bad universality”, local and not general?

L83: “climatic factors and mixed pixels” the hierarchy and nature of these should be separated (veg. function might be dependent on climate, but pixel mixing is an effect of the sensors spatial resolution).

261: “After experiments” the authors must clarify.

The methods section looks well explained, however over explanation in the 5 steps led to repetition.

Overall: this is an interesting paper. However, the authors must try to reduce its length.

Author Response

Dear Reviewer,

We thank your comments and suggestions that are very helpful for improving our manuscript (remotesensing-494934). At the same time, we are very encouraged by your affirmation to the study. We appreciate your responsible attitude toward our paper, as well as your impressive comments. We have made a big revision to the manuscript according to your comments. Our item-by-item responses to your comments are in the PDF file, where our responses are in red fonts. The major modifications of the paper are marked by blue background.

Yours sincerely,

Reviewer 3 Report

The paper is already in good shape, with a clear logic and a sound depiction of the methods and results. The proposed method looks very interesting and useful, as shown in Figs. 3-6.

I have several minor suggestions for the authors to further improve their work.

1)      some of the sentences in section 2.1 can be moved into the first paragraph in Introduction. This will help the potential readers get a better understanding of your study.

2)      Line 47, references for the human activities statement?

3)      How did you process the Landsat data exactly? Has an atmosphere correction been performed? Or is atmosphere correction necessary in the sanjiangyuan region? I feel more details or discussion should be provided here.

4)      Line 250, after experimentS.

5)      Line 262, VImax was typed twice here.

6)      Line 283-284, any explanation for the threshold of 28? Also, for the threshold of 0.2 in lines 249-252, more details would be much better. These two thresholds seem critical for the following analyses. Could the authors provide some suggestions for the potential readers on how to choose these two thresholds in other regions?

7)      Change the figure legends of continuous color into classified/discrete colors would be better. For example, in Fig. 3c, the green color means a value of 10, 20 or 50?  From Fig. 4 and Fig. 5, I can tell maybe the difference is around 10, but it will be better if Fig. 3 can show the spatial pattern.

8)      Line 327, maybe a naïve question: how the Zhang method was implemented? Is there a dataset with Zhang method available for downloading?

9)      Line 329, the Zhang method was shown in Fig. 3b, right?

Author Response

Dear Reviewer,

We thank your comments and suggestions that are very helpful for improving our manuscript (remotesensing-418489). At the same time, we are very encouraged by your affirmation to the study. We appreciate your responsible attitude toward our paper, as well as your impressive comments. We have revised our manuscript accordingly. Our item-by-item response to your comments is as following, where our response is in red fonts in the PDF file. The major modifications of the paper are marked by yellow background.

Yours sincerely,

Round 2

Reviewer 1 Report

I would like to thank the authors for the thorough and careful work they did to answer my comments on the first draft of this manuscript. I have only a few comments on their revision.

In my original comment about line 258 I wrote that green-up was located at the "maximum curvature" (the maximum in the second derivatrive) not the "maximum rate of change in curvature". The authors made this change at line 258 but not where this phrase occurs elsewhere in the manuscript. In manuscript version 3 I still see references to the "maximum rate of change in curvature" in lines 20, 69, and 212.

In the Abstract and Conclusions they write:

"Statistical analysis shows that the green-up date is significant relation to elevation gradient. And the green-up date delays obviously along with the elevation."

Better to say something like:

"Regression analysis shows that the green-up date is correlated to elevation: the green-up date is clearly later a higher elevations."

Section 3.3.1. Examining the data in Figure 6, I don't see a major break in the year 2006. I see that greenup in 2000 was quite early but otherwise the trend has been generally down since 2001. It is good that the authors put the p-values on this figure, they show that actually none of the trends are highly significant (if we accept p = 0.05 as the threshold of highly significant). In other words, there is weak statistical support for a change in the trend of green-up in 2006. Unless you have a strong hypothesis about some kind of climatic regime change in 2006, I don't see a good reason to emphasize a break in 2006. Likewise, the slopes reported in Figure 7a probably don't represent some kind of linear upward trend of 1 day per year, they are just the result of one very early year (2000) at the start of the 2000-2006 period. Most of them are probably non-significant. The same applies to Fig. 7b.

I think I would do one of the following with the 2000-2016 trend data:

1) Simply present Fig. 6 and report that the trend is weak due to considerable year-to-year variation; omit Fig 7.

2) Try a Mann-Kendall test on the data in Fig. 6 for 2000-2016. It may show better significance, though it is often quite similar to linear regression.

3) Examine the significance (p-value) of linear regression and/or Mann-Kendall test on a pixel-by=pixel basis for the whole study area, 2000-2016. Make this a new Fig. 7. The same software that they used to produce Fig. 7 should be able to generate a raster of the p-values also. This analysis will probably show that for most of the study area, the trend was weak for 2000-2016, but there may be parts of the study area where the trend was significant. In general though, I think the MODIS record is just a little too short to see a strong trend in the start of green-up, because of the year-to-year variation. Other researchers have found similar results (e.g. Remote Sens. 2017, 9, 514; doi:10.3390/rs9060514). Any of these changes would require corresponding revisions to the Conclusions and Abstract.

Author Response

Dear Reviewer,

We thank your comments and suggestions that are very helpful for improving our manuscript (remotesensing-494934). At the same time, we are very encouraged by your affirmation to the last revised version. We appreciate your responsible attitude toward our paper, as well as your impressive comments. And we tried the methods you mentioned, and finally decided the final solution. We have made a big revision to the manuscript according to your comments. We have learned a lot from your comments about the attitude of doing research and writing articles. Our item-by-item responses to your comments are as following, where our responses are in red fonts. The major modifications of the paper are marked by gray background.

Yours sincerely,

*************************************************************************************

Comments and Suggestions for Authors

Main Comments

I would like to thank the authors for the thorough and careful work they did to answer my comments on the first draft of this manuscript. I have only a few comments on their revision.

In my original comment about line 258 I wrote that green-up was located at the "maximum curvature" (the maximum in the second derivatrive) not the "maximum rate of change in curvature". The authors made this change at line 258 but not where this phrase occurs elsewhere in the manuscript. In manuscript version 3 I still see references to the "maximum rate of change in curvature" in lines 20, 69, and 212.

Response: Thanks for your comments. Please forgive our omissions. For the problem you mentioned, we have revised the full text and revised the lines 20, 69, 212 and 480.

In the Abstract and Conclusions they write:

"Statistical analysis shows that the green-up date is significant relation to elevation gradient. And the green-up date delays obviously along with the elevation."

Better to say something like:

"Regression analysis shows that the green-up date is correlated to elevation: the green-up date is clearly later higher elevations."

Response: Thanks for your helpful comments. We have modified this expression in the Abstract and Conclusion in lines 33-35 and 555-557 Respectively.

Section 3.3.1. Examining the data in Figure 6, I don't see a major break in the year 2006. I see that greenup in 2000 was quite early but otherwise the trend has been generally down since 2001. It is good that the authors put the p-values on this figure, they show that actually none of the trends are highly significant (if we accept p = 0.05 as the threshold of highly significant). In other words, there is weak statistical support for a change in the trend of green-up in 2006. Unless you have a strong hypothesis about some kind of climatic regime change in 2006, I don't see a good reason to emphasize a break in 2006. Likewise, the slopes reported in Figure 7a probably don't represent some kind of linear upward trend of 1 day per year, they are just the result of one very early year (2000) at the start of the 2000-2006 period. Most of them are probably non-significant. The same applies to Fig. 7b.

I think I would do one of the following with the 2000-2016 trend data:

1) Simply present Fig. 6 and report that the trend is weak due to considerable year-to-year variation; omit Fig 7.

2) Try a Mann-Kendall test on the data in Fig. 6 for 2000-2016. It may show better significance, though it is often quite similar to linear regression.

3) Examine the significance (p-value) of linear regression and/or Mann-Kendall test on a pixel-by=pixel basis for the whole study area, 2000-2016. Make this a new Fig. 7. The same software that they used to produce Fig. 7 should be able to generate a raster of the p-values also. This analysis will probably show that for most of the study area, the trend was weak for 2000-2016, but there may be parts of the study area where the trend was significant. In general though, I think the MODIS record is just a little too short to see a strong trend in the start of green-up, because of the year-to-year variation. Other researchers have found similar results (e.g. Remote Sens. 2017, 9, 514; doi:10.3390/rs9060514). Any of these changes would require corresponding revisions to the Conclusions and Abstract.

Response: Thanks for your constructive suggestions. Honestly, the change in vegetation green-up date is not obvious from 2000 to 2016, and it is weakly significant before and after 2006. So there is no major discovery in piecewise linear analysis. The short time sequence is still a defect, but this will be a place for follow-up research to improve.

We chose the third option for the three scenarios you gave. Because the Mann-Kendall method has bad applicability to data with short time series, we use this method to test the trend and significance of the vegetation green-up date from 2000 to 2016. As you have predicted, although the vegetation green-up date is generally weakly significant in time series, still 10.2% of the whole study area showed significant trend of advance. And these areas are mainly distributed in eastern, western and central Sanjiangyuan area.

Therefore, we modify the content of the article in the time series of the vegetation green-up date in the Sanjiangyuan area. We modified the trend test of vegetation green-up date in Section 3.3.1 in lines 420-438. Finally, we revised the Abstract in lines 31-34, Conclusion in lines 551-554, and Methods in lines 323-324.

This manuscript is a resubmission of an earlier submission. The following is a list of the peer review reports and author responses from that submission.

Round 1

Reviewer 1 Report

I am happy to see this revised version of the manuscript. 

Author Response

Dear Reviewer,

We thank your comments and suggestions that are very helpful for improving our manuscript (remotesensing-447439). We appreciate your responsible attitude toward our paper. 

The major modifications of the paper are marked by green background.

Reviewer 2 Report

This paper proposed a method called GUD-CMAPD that integrate the widely used double logistic regression and polynomial fit and climate data to derive spring phenology in the Sanjiangyuan Nature Reserve. However, there are several issues in this paper that needs to be addressed. First, the idea is not original enough (both methods are already widely used in the currently literature). They simply use a threshold to determine which method to use. Second, the results were not validated against any other independent resources such as ground validation. The proposed RMSE only suggest the goodness of fit of the curve, not the GUD itself. Third, they compared their spatially smoothed products directly with Zhang’s method (MODIS phenology product, which are not spatially smoothed), which has some issues. Fourth, only The GUD was supposed to more correlated with growing degree days based on a threshold temperature (GDD) in the currently literature, however, the author simply used accumulated temperature. The specific comments are in the following:

Major comments:

1. I am very confused about 2.3.2 and 2.3.3. 2.3.2 is the method to calculate GUD using curvature (second derivation), then 2.3.3 is using another threshold method, I wonder did the author use two methods to derive GUD? Or did the author use the method in 2.3.2 as the initial input for outlier detection?

2. I don’t think it is correct to do a spatial outlier correction in section 2.3.3. The robustness of the curve strongly depends on the time series of NDVI at each pixel. If there are less number of quality data, the accuracy of the curve might be influenced. There is a strong spatial autocorrelation in the 8-neighborhood, doing a spatial outlier correction will only smooth the image, not improving the accuracy or improving the data quality.

3. Spring temperature is an important factor, but it is not the only factor. In fact, growing degree days (GDD) based on a threshold temperature (base temperature) is the driver. The authors may consider using GDD instead of using accumulated temperature. In addition, the spatial resolution of the temperature will also affect the robustness of the method because MODIS data is 500-meter

4. Section 3.1. I don’t think the author should directly compare Zhang’s method with the prosed GUD-CMAPD. The GUD-CMAPD smoothed the data but Zhang’s did not, therefore, the proposed methods looks smoother (or more continuous) than Zhang’s.

5. Figure 5, again, the Zhang’s method did not smooth the data or use climate data.

6. Table 2: again, RMSE only shows goodness of fit of the curve, it did not indicate the accuracy of derived GUD from the curve. The author might consider using some ground observations.

Minor comments:

Line 144: the MODIS data described above have a 500m spatial resolution à which data did the author refer to? Because MYD09A1 has 8-day temporal resolution.

Line 178: please clearly described how the 30-meter data was resampled to 500 meter. Did you use the major land cover types within each 500m pixel?

Line 219: why use degree of 5 polynomial fitting? In my opinion, unlike logistics fit (five parameter indicates background, growth rate, timing of change point, and maximum values), polynomial fit has no physical meanings? Why did the author use polynomial?

Line 222-223: in the figure 2, the maximum value threshold was set to 0.4, but those two lines indicate 0.2, which one did the authors use? If 0.2 was used, why use this threshold?

Line 224: This RMSE did not indicate the method is robust enough to derive vegetation phenology, it only provide an index of “goodness of fit”. The authors should use gorund observations or other independent observations to validate their methods.

Line 263: reference 33 is about fall phenology, not spring phenology.

Author Response

Dear Reviewer,

We thank your comments and suggestions that are very helpful for improving our manuscript (remotesensing-447439). We appreciate your responsible attitude toward our paper, as well as your impressive comments. We have revised our manuscript accordingly. Our item-by-item response to your comments is in the PDF file, where our response is in red fonts. The major modifications of the paper are marked by green background.

Reviewer 3 Report

1)    Please provide reference for Lines 65-66 “However, the method has low precision over large areas and areas with low vegetation coverage”.

2)    In line 77-78: The formation of sentence is strange.

Particularly “… the threshold of curvature change….. maximum value of vegetation index” is confusing. Do the authors mean to say threshold and rate of change of curvature as mentioned earlier in lines 63-68.  Please explain.

Additionally, please clarify how the “threshold of curvature change” is affected by the maximum value of vegetation index.

3)    Line 123-124: The authors mention that the MODIS products include Landsat-7 data. Please verify and correct.

4)     In Lines 152-161, Landsat 8 data is described. Do the authors use both Landsat-7 and Landsat-8 to cover the time period of study i.e. 2000-2016? How was the SLC error in landsat 7 corrected? In its present state section 2.2.1 is confusing and needs clear details of the various data used and the time period under study.

5)    In line 180: please describe the “eliminate the abnormal pixels during the reincarnation period.” What are abnormal pixels?

6)    Please clarify the flowchart in Figure 2.:

The authors select the best time series between MODIS and Landsat data for a pixel. This selection is mentioned only in the flow chart and needs to explained in the body of the paper too. How was this selection done? and how was Landsat resampled to match the spatial resolution of different MODIS products?

Additionally, in Step1 of flowchart there is a conditional operator for NDVImax>0.4, whereas, in lines 222-223 authors explain NDVImax>0.2 and through out the paper. Please explain this discrepancy.

7)    The differences in the estimated phenology green-up date is described in lines 216-217 and figure 3. I think a difference map between 3a and 3b would be better.

8)    In line 278 the abbreviation GUD-CMAPD is mentioned for the first time. Introducing GUD-CMAPD clearly would help improve the readability of the manuscript.

9)    Line 290: GUD-PL is not described earlier.

10)  Till section 4.2 the paper is mostly about NDVI and then EVI is suddenly introduced. Please clarify the choice of data products from the beginning of the paper.

Author Response

Dear Reviewer,

We thank your comments and suggestions that are very helpful for improving our manuscript (remotesensing-447439). We appreciate your responsible attitude toward our paper, as well as your impressive comments. Our item-by-item response to your comments is in the PDF file, where our response is in red fonts. The major modifications of the paper are marked by green background.

Round 2

Reviewer 3 Report

A few minor changes are still required in the manuscript.

Some errors that I could still identify are (but not limited to):

1) Figure 4c. is lacking a legend

2) line 67, I am not sure if the thresholds G20 and G50 can be called "growth rate". The term "ratio" might be more appropriate.

 3) line 368, it should be figure 4d not 4c.

Author Response

Dear Reviewer,

Thanks for your comments and suggestions on our manuscript entitled “Improved Vegetation Spring Phenology Calculation Method using a Coupled Model and Abnormal Point Detection” (remotesensing-447439). Your suggestions are very helpful, and your hard work was appreciated deeply. We have carefully revised the entire manuscript. We modified the manuscript according to your suggestions. Besides, we have also revised the references, words and other aspects. Our item-by-item response to your comments is as following. The major modifications of the paper are marked by blue background.

Response to Reviewer Comments

A few minor changes are still required in the manuscript.

Some errors that I could still identify are (but not limited to):

1)      Figure 4c. is lacking a legend

Response: Thanks for your helpful suggestion. Based on your suggestion, we have modified Figure 4(d). The legend is added to Figure 4(d), and Figure 4(a), (b) and (c) is modified to uniformity and aesthetics of the figure, shown in lines 337-338.

2)      line 67, I am not sure if the thresholds G20 and G50 can be called "growth rate". The term "ratio" might be more appropriate.

Response: Thanks for your comments. We verified White et al. ‘s paper [1], and the "ratio" was used in the paper which proposed this theory, so we revised it in the article, shown in line 67.

 3) line 368, it should be figure 4d not 4c.

Response: Please forgive our negligence. We have fixed this error, as shown in line 357.

Reference

1.     White, M.A.; Thornton, P.E.; Running, S.W. A continental phenology model for monitoring vegetation responses to interannual climatic variability. Global biogeochemical cycles 1997, 11, 217–234.